# Protein arginine methyltransferase 2 controls inflammatory signaling in acute myeloid leukemia
Camille Sauter [1] ✉, Thomas Morin[1], Fabien Guidez[1], John Simonet[1], Cyril Fournier[1,2], Céline Row[1,2,3], Denis Masnikov[1], Baptiste Pernon[1], Anne Largeot[1,8], Aziza Aznague[1,4], Yann Hérault [5], Guy Sauvageau[6], Marc Maynadié[1,3], Mary Callanan[1,2,4], Jean-Noël Bastie[1,7], Romain Aucagne[1,2,9] & Laurent Delva [1,9] ✉

Arginine methylation is catalyzed by protein arginine methyltransferases (PRMTs) and is involved in various cellular processes, including cancer development. PRMT2 expression is increased in several cancer types although its role in acute myeloid leukemia (AML) remains unknown. Here, we investigate the role of PRMT2 in a cohort of patients with AML, PRMT2 knockout AML cell lines as well as a *Prmt2* knockout mouse model. In patients, low *PRMT2* expressors are enriched for inflammatory signatures, including the NF-κB pathway, and show inferior survival. In keeping with a role for PRMT2 in control of inflammatory signaling, bone marrow-derived macrophages from *Prmt2* KO mice display increased pro-inflammatory cytokine signaling upon LPS treatment. In PRMT2-depleted AML cell lines, aberrant inflammatory signaling has been linked to overproduction of IL6, resulting from a deregulation of the NF-κB signaling pathway, therefore leading to hyperactivation of STAT3. Together, these findings identify PRMT2 as a key regulator of inflammation in AML.

Arginine methylation of proteins is involved in the regulation of various cellular processes, including DNA repair, gene transcription and post-transcriptional regulation[1], and is catalyzed predominantly by protein arginine methyltransferases (PRMTs), which transfer a methyl group from S-adenosyl-methionine to the guanidine nitrogen on an arginine residue of the partner protein[2]. In addition to PRMTs, two other arginine methyltransferases have been identified in mammals: NDUFAF7 and METTL23[3]. PRMTs are classified into three types based on their enzymatic activity: type I PRMTs (1, 2, 3, 4, 6, and 8) form ω-$N^G$-monomethylarginine (MMA) and ω-$N^G$,$N^G$-asymmetric dimethylarginine (ADMA), whereas type II PRMTs (5 and 9) catalyze the production of MMA and ω-$N^G$,$N^G$-symmetric dimethylarginine (SDMA). Finally, PRMT7, as the single type III, only produces MMA[4]. In hematological malignancies, PRMTs are major actors in many cellular mechanisms, e.g. proliferation, cell cycle, inhibition of apoptosis, and RNA splicing[5]. Moreover, PRMTs are also involved in controlling gene expression through methylation of arginine residues on histone tails. There are currently phase I/II clinical trials using PRMT inhibitors showing promising results on hematological malignancies, especially acute myeloid leukemia (AML), lymphomas, and myelodysplastic neoplasms[6].

Among the nine members of the family, PRMT2 is the only one harboring a Src Homology 3 (SH3) binding domain, which is essential for its catalytic activity[7]. Although several partners of PRMT2 have been described, its methylation substrates remain widely unknown[8]. In cancer, *PRMT2* expression is upregulated in hepatocellular carcinoma and glioblastoma, and is involved in tumor development[9,10]. Conversely, its expression is decreased in cardia gastric cancer[11], suggesting potential antitumor activity. We previously identified PRMT2 among other binding partners of the lysine acetyltransferase KAT6A (MOZ) through a double-hybrid screening. As KAT6A is crucial for hematopoiesis and involved in many chromosomal translocations found in AML, it is of high importance to define the precise roles of PRMT2. Indeed, this arginine methyltransferase could have non-

[1]Inserm UMR 1231, Epi2THM team, LabEx LipSTIC Team, UFR des Sciences de Santé, Université de Bourgogne, Dijon, France. [2]Unit for Innovation in Genetics and Epigenetics in Oncology, Dijon University Hospital, Dijon, France. [3]Department of Hematology Biology, University Hospital Dijon Bourgogne François-Mitterrand, Dijon, France. [4]Inserm UMS 58 BioSanD, CRISPR Functional Genomics (CRIGEN) facility, UFR des Sciences de Santé, Université de Bourgogne, Dijon, France. [5]Université de Strasbourg, CNRS UMR7104, Inserm U1258, Institut de Génétique et de Biologie Moléculaire et Cellulaire (IGBMC), Illkirch-Graffenstaden, France. [6]Molecular Genetics of Stem Cells, Institute for Research in Immunology and Cancer (IRIC), Université de Montréal, Montréal, QC, Canada. [7]Department of Clinical Hematology, University Hospital Dijon Bourgogne François-Mitterrand, Dijon, France. [8]Present address: Tumor Stroma Interactions, Department of Oncology, Luxembourg Institute of Health, Luxembourg, Luxembourg. [9]These authors jointly supervised this work: Romain Aucagne, Laurent Delva. ✉e-mail: camille.sauter@u-bourgogne.fr; laurent.delva@u-bourgogne.fr

redundant functions from the other PRMTs, especially the other type I, which have common methylation marks.

To determine the role of PRMT2 in hematopoiesis and AML, we use a murine knockout (KO) model (*Prmt2*⁻/⁻ mice), KO AML cell lines (*PRMT2*ᴷᴼ), as well as patients with AML clinical and RNA sequencing (RNA-seq) data. We demonstrate that PRMT2 controls the inflammatory signaling in mouse and human hematopoiesis. Specifically, our work establishes that PRMT2 controls STAT3 activation, possibly through the downregulation of the NF-κB signaling pathway in AML cells, inducing inhibition of IL6 production. In the same manner, patients with AML harboring low expression of *PRMT2* display a high inflammatory signature associated with an increased expression of several NF-κB-related genes and may exhibit a lower survival rate. Altogether, our results identify PRMT2 as a key regulator of the inflammatory phenotype in AML.

## Results

### PRMT2 modulates the inflammatory phenotype of patients with AML

KAT6A (lysine acetyltransferase 6 A), formerly called MOZ or MYST3, is involved in normal hematopoietic stem/progenitor cells functions. We have previously demonstrated that KAT6A is a partner of KMT2A (MLL) and Symplekin in hematopoietic stem/progenitor cells[12,13]. Compared to other MYST family members, KAT6A and KAT6B (or MORF) harbor unique structural functional domains[14]. Meanwhile, KAT6A contains a specific PQ (Pro/Gln-stretch) insertion in its SM (Ser/Met-rich) domain (Supplementary Fig. 1a). To establish new hematopoietic-specific binding partners of KAT6A, a yeast two-hybrid screening has been performed with the KAT6A-specific PQ domain as bait and a human BM cDNA library as prey (Supplementary Fig. 1b). This screening reveals PRMT2 as one of the potential interaction partners of KAT6A, and we further confirmed the possible interaction by co-immunoprecipitation in human embryonic kidney (HEK293) cells (Supplementary Fig. 1c). Knowing the importance of KAT6A fusion proteins in AML initiation and maintenance, we decide to focus our attention on the role of its partner PRMT2 in this context.

Several studies have uncovered differentially expressed PRMTs across various types of cancers, and it has been reported that PRMTs are deregulated in hematological malignancies, i.e. lymphoma and leukemia[6]. Thus, to define the extent of PRMT2 involvement in AML initiation and maintenance, we analyzed *PRMT2* expression in 371 patients with de novo AML (excluding acute promyelocytic leukemias) (Fig. 1a) using data from The Leucegene Project (Table 1). In a univariate analysis, survival curves of patients with a *PRMT2* expression above and below the median expression of the cohort are compared and reveal that patients with a reduced *PRMT2* expression display a lower survival rate (Fig. 1b). Reanalysis of exome sequencing data from 105 patients within the same cohort, to score for mutations of known AML genes, does not reveal differences between high vs low *PRMT2* expression groups either in deciles ($n = 10$, Fig. 1c) or median threshold ($n = 52$, Supplementary Fig. 2a), thus indicating that poor survival across high versus low *PRMT2* groups reflects PRMT2-linked signaling processes.

To take the analysis further, we take advantage of the RNA-seq data to divide these patients with AML into two groups (by deciles), showing high (90% decile) and low *PRMT2* (10% decile) expression levels (*PRMT2*ˡᵒʷ, $n = 37$ and *PRMT2*ʰⁱᵍʰ, $n = 37$; Table 1: patient characteristics), respectively. We perform a gene set enrichment analysis (GSEA) across these two groups. This reveals 11 gene sets (FDR < 25%, $p < 0.01$) that are significantly enriched in *PRMT2*ˡᵒʷ compared to *PRMT2*ʰⁱᵍʰ patients with AML. Predominant among these gene signatures is a TNF via NF-KB signaling hallmark (Fig. 1d, e). We find that expressions of *NFKB1*, *NFKB2*, *REL*, *RELA* and *RELB*, coding for NF-κB functional subunits, are significantly increased in the *PRMT2*ˡᵒʷ patients (Fig. 1f). We confirm that these same genes behave in the same way in the *PRMT2*ˡᵒʷ group of patients ($n = 30$) of another cohort called BeatAML ($n = 302$) (Supplementary Fig. 2b). Moreover, a GSEA comparing the BeatAML *PRMTs*ˡᵒʷ versus *PRMT2*ʰⁱᵍʰ patients

also shows the TNF via NF-KB signaling hallmark as the top enriched set of genes in *PRMT2*ˡᵒʷ patients (Supplementary Fig. 2c).

Within the Leucegene Project, we find that the *PRMT2*ˡᵒʷ group of patients also displays significantly higher expressions of the *IL6* and *TNF* pro-inflammatory cytokines (Fig. 2a, b). Recent studies have shown that serum ferritin is deregulated in most patients with AML, and high levels of ferritin are correlated with inflammation and chemoresistance[15]. Our analysis reveals that *PRMT2*ˡᵒʷ patients showing an exacerbated inflammatory signature also display higher expressions of *FTH1* and *FTL* genes, corresponding to genes coding for heavy and light chains of ferritin, respectively (Fig. 2c, d). Moreover, *FTH1* and *FTL* expressions are both inversely correlated with *PRMT2* expression within the patients with AML (Supplementary Fig. 2d, e).

Since inflammatory signaling can remodel the immune landscape, we analyze the immune cell composition of the AML samples by using ImmuCellAI analysis (Immune Cell Abundance Identifier), a gene set signature-based method[16]. *PRMT2*ˡᵒʷ patients display a significantly higher frequency of monocytes and a decreased abundance of NKT cells compared to *PRMT2*ʰⁱᵍʰ patients (Fig. 2e, f), suggesting an increased activation of the innate immune response and potential deregulation in the development and maintenance of these cells, as well as a possible influence of PRMT2 on the expression or activity of key regulators involved in monocyte and NKT cell development, recruitment, or survival. Altogether, these findings indicate that PRMT2 could be involved in the molecular mechanism of inflammatory regulation in AML, thus explaining the increased expression of pro-inflammatory genes in patients displaying low *PRMT2* expression.

### Inflammation pathways are enriched in absence of PRMT2 in AML cells

To assess the roles of PRMT2 in AML, we generate *PRMT2* knockout cell lines (*PRMT2*ᴷᴼ) from the lipopolysaccharide (LPS)-sensitive HL-60 cells (Fig. 3a). It has previously been shown that the loss of activity of one PRMT, through inhibition of activity or expression, can be compensated by other PRMTs through so-called substrate scavenging[17,18].

Assuming a possible compensatory role for PRMTs, we therefore investigate whether PRMT2 KO causes upregulation of the expression of other PRMTs at the protein level. Our finding indicates that the protein expression level of other PRMTs remains stable in KO cells compared to PRMT2 WT (Fig. 3a). Additionally, no significant effect of PRMT2 depletion on cell proliferation (Fig. 3b) or on myeloid cell surface markers is found (Supplementary Fig. 3).

We carried out RNA-seq to determine the possible effects of PRMT2 on gene expression in the WT and KO HL-60 cells. GSEA comparing *PRMT2*ᴷᴼ to WT cells reveals an enrichment of 7 sets of genes (FDR < 25%, $p < 0.01$). Interestingly, we find the genes involved in the NF-κB signaling pathway in response to TNF as the most enriched set of genes, thus confirming what we previously showed in the *PRMT2*ˡᵒʷ patients with AML (Fig. 3c, d). In keeping with observations in the data from patients with AML, we highlight NF-κB subunit genes in the top 20 enriched genes in the "TNFA signaling via NFKB" hallmark set of genes in *PRMT2*ᴷᴼ cells (Fig. 3e). Moreover, the inflammation-related IL6/JAK/STAT3 signaling pathway is also enriched in the *PRMT2*ᴷᴼ HL-60 cells. Taken together, this is evidence of the role of PRMT2 in the control of inflammatory signaling in AML.

### *Prmt2*⁻/⁻ bone marrow-derived macrophages exhibit a dysregulated inflammatory response

To deepen our understanding of the effects of PRMT2 on inflammation, we use *Prmt2* knockout (*Prmt2*⁻/⁻) mice[19]. We investigate their hematopoietic compartment and reveal that PRMT2-deficient mice show no discernible variation in the populations of mature and immature blood and bone marrow (BM) cells (Supplementary Fig. 4a, b). Similarly, *Prmt2*⁻/⁻ BM cells demonstrate comparable hematopoietic reconstitution ability to normal BM cells when transplanted in a competitive setting (Supplementary Fig. 4c).

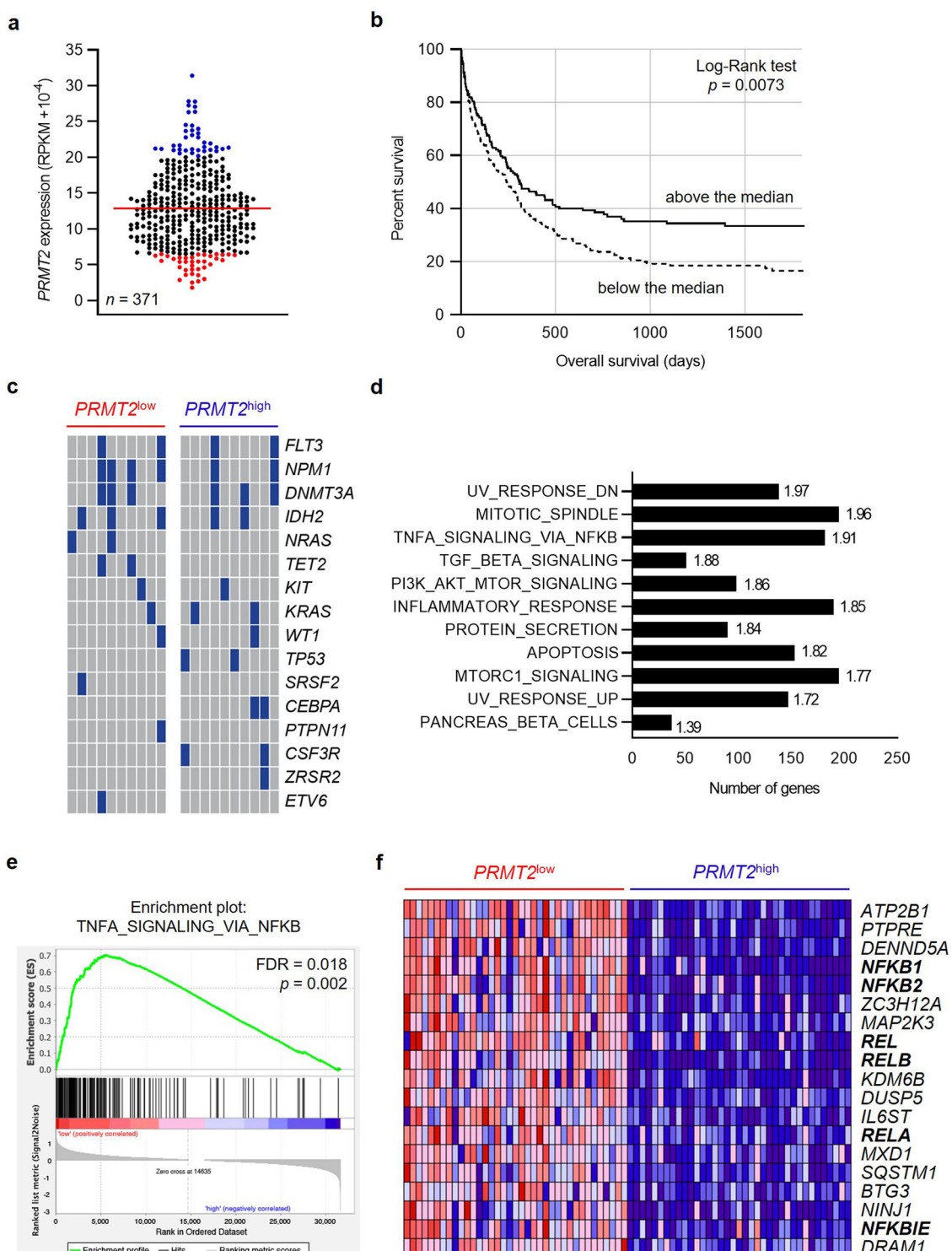

**Fig. 1 | Patients with AML displaying a low *PRMT2* expression show a higher inflammatory signature. a** *PRMT2* gene expression within the Leucegene cohort (non-APL AML, *n* = 371) at diagnosis with the *PRMT2^low^* patients in red (*n* = 37) and *PRMT2^high^* patients in blue (*n* = 37). **b** Kaplan-Meier survival curves of the patients depending on their *PRMT2* expression: above or below the median of expression of the cohort. **c** Mutation profile of the *PRMT2^low^* (*n* = 10) and *PRMT2^high^* (*n* = 10) patients for a subset of frequently mutated genes in AML. Blue: mutated gene; gray:

non mutated gene. **d** The bar graph shows the most significantly overrepresented hallmarks after a gene set enrichment analysis (GSEA) of *PRMT2^low^* compared to *PRMT2^high^* patients with normalized enrichment scores on the right. All gene sets are FDR < 25% and *p* < 0.01. **e** GSEA plot for the "TNFA signaling via NFKB" hallmark with FDR and nominal *p* value indicated. **f** Heatmap representing the top 20 enriched genes of the "TNFA signaling via NFKB" hallmark. NF-κB-related genes appear in bold.

**Table 1 | Clinical characteristics of the patients from the Leucegene cohort, with the PRMT2$^{low}$ and PRMT2$^{high}$ groups**

| | Leucegene (n = 371) | PRMT2$^{low}$ (n = 37) | PRMT2$^{high}$ (n = 37) | Statistical significance |
|---|---|---|---|---|
| De novo | 346 (93.3%) | 36 (97.3%) | 35 (94.6%) | |
| t-AML | 25 (6.7%) | 1 (2.7%) | 2 (5.4%) | |
| Sex | | | | |
| Male | 210 (56.6%) | 19 (51.4%) | 23 (62.2%) | |
| Female | 161 (43.4%) | 18 (48.6%) | 14 (37.8%) | |
| Age (years) | | | | |
| Median | 58 | 57 | 55 | |
| Range | 17-87 | 22-82 | 18-84 | |
| WBC count (x10$^9$/L) | | | | |
| Median | 33.2 | 14.5 | 73.4 | *** |
| Range | 0.8-361.2 | 1.3-322.5 | 2.9-361.2 | |
| Cytogenetic risk | | | | |
| Favorable | 46 (12.4%) | 4 (10.8%) | 9 (24.3%) | |
| Intermediate | 206 (55.5%) | 19 (51.4%) | 22 (59.5%) | |
| Adverse | 117 (31.5%) | 14 (37.8%) | 6 (16.2%) | |
| FAB subtype | | | | |
| M0 | 23 (6.2%) | 2 (5.4%) | 4 (10.8%) | |
| M1 | 103 (27.8%) | 10 (27.0%) | 14 (37.8%) | |
| M2 | 50 (13.5%) | 9 (24.3%) | 7 (18.9%) | |
| M4 | 56 (15.1%) | 4 (10.8%) | 4 (10.8%) | |
| M5 | 63 (17%) | 5 (13.5%) | 5 (13.5%) | |
| M6 | 8 (2.2%) | 1 (2.7%) | 0 | |
| M7 | 2 (0.5%) | 0 | 0 | |
| Not classificable | 66 (17.8%) | 6 (16.2%) | 3 (8.1%) | |

*WBC* white blood count.

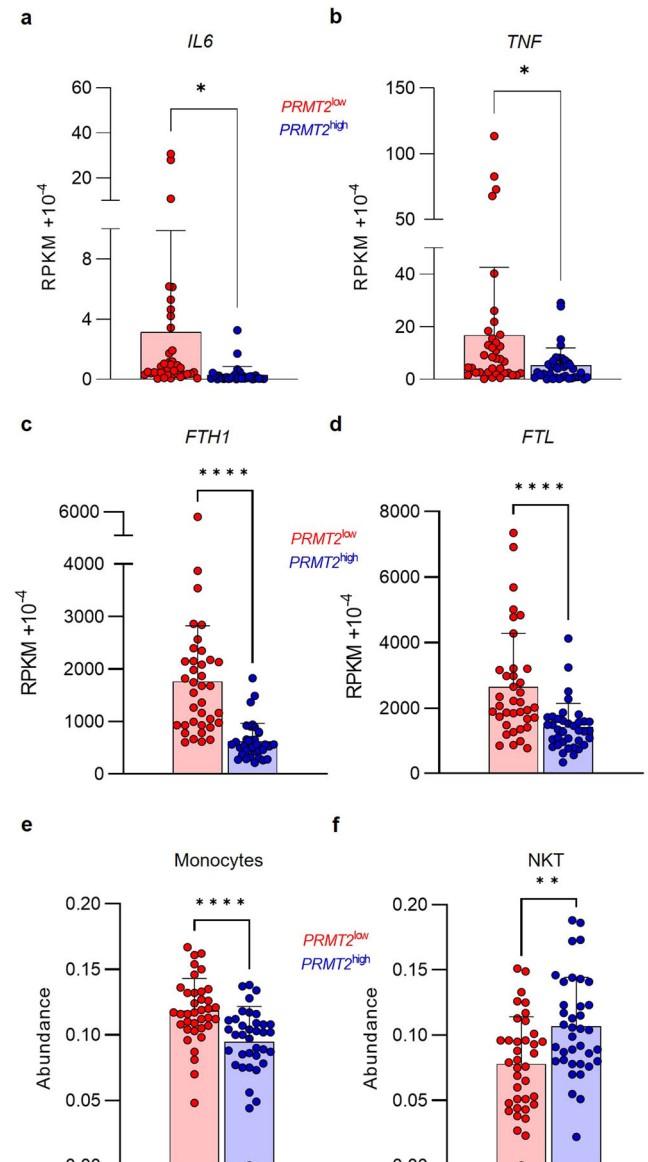

**Fig. 2 | PRMT2 is associated with inflammatory factors' expression.** Gene expression of *IL6* (**a**), *TNF* (**b**), *FTH1* (**c**) and *FTL* (**d**) within the *PRMT2$^{low}$* patients in red (*n* = 37) and *PRMT2$^{high}$* patients in blue (*n* = 37). ImmuCellAI analysis showing monocyte (**e**) and NKT cell (**f**) abundance within the *PRMT2$^{low}$* patients in red (*n* = 37) and *PRMT2$^{high}$* patients in blue (*n* = 37). Error bars represent the mean ± SD. Paired *t* tests with *p* values indicated in the graphs: *$p < 0.05$; **$p < 0.01$; ****$p < 0.0001$.

Subsequently, we examine whether the inflammatory phenotype of LPS-activated BM-derived macrophages (BMDMs) depends on PRMT2 expression. To this end, we generate an ex vivo model of BMDMs using monocytes isolated from the BM of *Prmt2$^{-/-}$* and control mice (Fig. 4a), to obtain a homogenous population of the immune system. An inflammatory phenotype is triggered using LPS following 5 days of macrophage differentiation. *Prmt2$^{-/-}$* macrophages do not reveal any morphological change (Fig. 4b, Supplementary Fig. 4d) or any variation in the expression of surface markers either with or without LPS treatment (Supplementary Fig. 4e).

However, stimulation of *Prmt2$^{-/-}$* BMDMs by LPS causes a significantly higher expression of *IL6* and *TNF* pro-inflammatory cytokines (Fig. 4c, d), and it also results in an increase in their secretion in the culture medium (Fig. 4e, f). This demonstrates an enhanced response to LPS in the absence of PRMT2, thus confirming a role for PRMT2 in the control of the inflammatory phenotype in mouse BMDMs, especially an interesting increase in IL6 production and secretion.

Taken together, these data suggest that even though PRMT2 does not seem to have a crucial role in the formation of mature blood cells in homeostatic conditions, it alleviates the inflammatory response induced by LPS in the mouse.

## PRMT2 is involved in the regulation of STAT3 activation

The HL-60 cell line can differentiate into various cell types in vitro and is also sensitive to LPS, which triggers an inflammatory phenotype in these cells[20,21]. To test the findings from the mice in human AML cells, especially through the IL6-related pathways, we have treated *PRMT2$^{KO}$* HL-60 cells with LPS. This produces higher levels of *IL6* expression and secretion compared to

WT AML cells (Fig. 5a, b). Since it is known that the binding of IL6 to its receptor (IL6R) contributes to the activation of JAK/STAT signaling pathway, we measure the IL6R expression in *PRMT2$^{KO}$* cells (Supplementary Fig. 5a) and we do not find any difference. Hence, our KO cells provide an opportunity to investigate mechanisms driving enrichment of IL6/JAK/STAT3 hallmark signatures as we detected in the GSEA in *PRMT2$^{KO}$* cells compared to control.

In the absence of PRMT2 and under LPS stimulation, STAT3 activation, characterized by its phosphorylation at tyrosine 705 (Y705), is highly increased compared to control cells, without any significant change in STAT3 protein expression (Fig. 5c, d). By performing a rescue experiment, we confirm that the observed phenotype is due to the specific loss of PRMT2. Indeed, levels of phosphorylated STAT3 are reduced in *PRMT2$^{KO}$* cells which are rescued by an overexpression of PRMT2 (Fig. 5e, f). Interestingly, the level of phosphorylated STAT3 is equivalent in the cells rescued

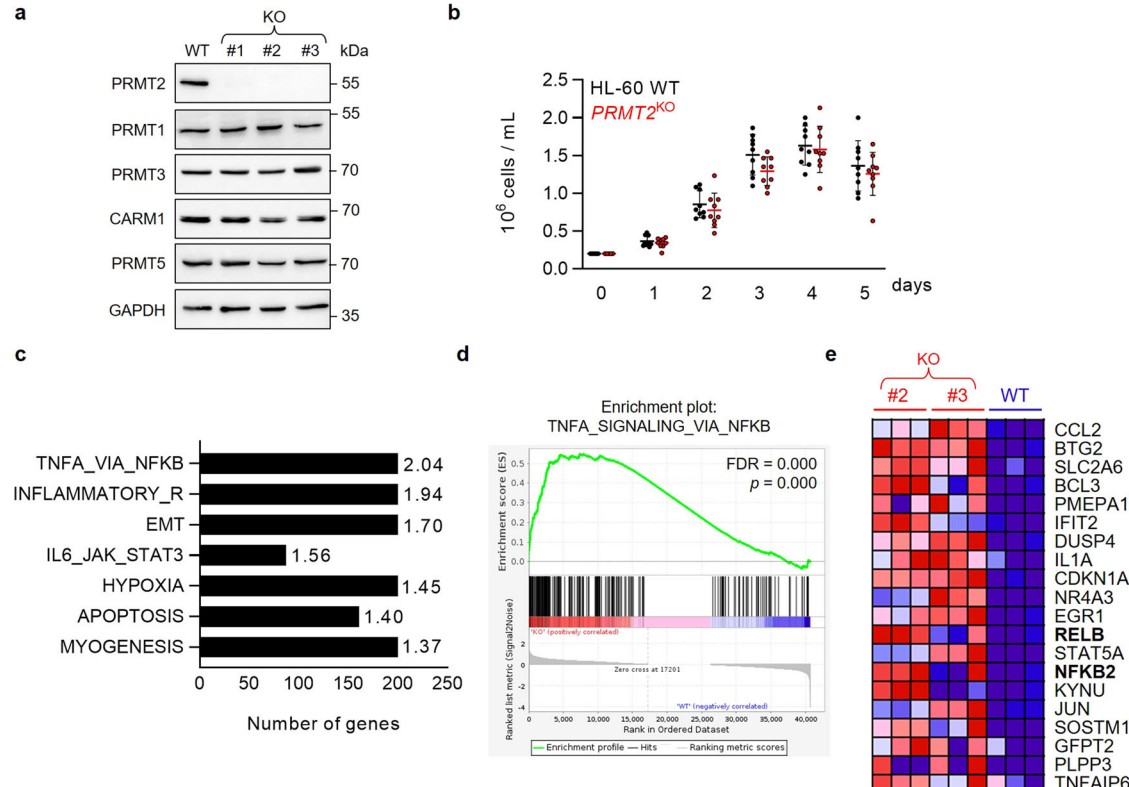

**Fig. 3 | The loss of PRMT2 increases the inflammatory phenotype of leukemic cells in vitro. a** Western Blot analysis on WT HL-60 or three different clones of *PRMT2*[KO] (KO) cells for PRMT2 and other PRMT (-1, -3, CARM1, and PRMT5) protein expressions. GAPDH is used as loading control. The presented blot is a representative figure from three independent experiments. **b** Assessment of cell proliferation of the 3 *PRMT2*[KO] clones compared to WT HL-60 cells by cell counting, shown as mean ± SD of three independent experiments (*n* = 9). **c** The bar graph represents the most significantly overrepresented hallmarks after a gene set enrichment analysis (GSEA) of #2 and #3 *PRMT2*[KO] clones compared to WT with normalized enrichment scores on the right. All gene sets are FDR < 25% and *p* < 0.01. **d** GSEA plot for the "TNFA signaling via NFKB" hallmark with FDR and nominal *p* indicated. **e** Heatmap representing the top 20 enriched genes of the "TNFA signaling via NFKB" hallmark. NF-κB-related genes appear in bold.

with a wild type and a catalytically dead mutant (E220Q) PRMT2 protein, indicating that the phosphorylation of STAT3 is not impacted by the loss of catalytic activity of PRMT2 (Fig. 5e). These results identify a regulatory role of PRMT2 on STAT3 after IL6 binding to its receptor, following LPS stimulation. Methylation of hypothalamic STAT3 by PRMT2 has already been described upon leptin stimulation[22]. We do not, however, identify any interaction between PRMT2 and STAT3 after co-immunoprecipitation assays in a model of overexpression of the citrine-PRMT2 fusion protein in HL-60 cells (Supplementary Fig. 5b).

Taken together, these findings suggest that STAT3 signaling is deregulated in absence of PRMT2. However, STAT3 is unlikely to be a substrate of PRMT2 in AML cells. Therefore, the modulation of STAT3 activity may arise from the scaffolding function of PRMT2.

### NF-κB-dependent activation of STAT3 is repressed by PRMT2

By treating HL-60 *PRMT2*[KO] cells with IL6 instead of LPS, the difference of phosphorylated STAT3 levels between the WT and *PRMT2*[KO] cells is abolished, meaning that only the LPS-triggered activation of STAT3 is impacted by the absence of PRMT2 (Fig. 6a, b). Other signaling pathways such as the MAPK pathways can lead to the activation of the STAT3 protein. Nevertheless, we do not find any difference between the activated and total form of the studied MAPKs, i.e. SAPK/JNK, ERK1/2, and p38 (Fig. 6c) in the *PRMT2*[KO] cells compared to WT AML cells, suggesting that PRMT2 has little to no role to play in these pathways upon LPS stress.

Under LPS stimulation of myeloblastic cells such as HL-60, IL6 production can be triggered by several signaling modules, including NF-κB[23,24]. Interestingly, following LPS stimulation in HL-60 *PRMT2*[KO] cells, increased levels of IL6 appear to result from an overactivated NF-κB pathway, as

evidenced by an increased translocation of the p65 NF-κB subunit into the nucleus in *PRMT2*[KO] cells, most visible after 8 and 24 h of LPS stimulation (Fig. 6d). We also demonstrate that the increased phosphorylated STAT3 protein can enter more into the nucleus of *PRMT2*[KO] cells compared to WT (Fig. 6d). These results indicate that PRMT2 controls the STAT3 protein activation through the modulation of IL6 production, probably coming from the regulation of the NF-κB signaling pathway.

### Discussion

Here, we establish that PRMT2 downregulation leads to an overactivation of the inflammatory processes in AML. Indeed, a direct link between inflammation and AML has been previously highlighted, indicating that inflammation plays a major role in myelosuppression, chemoresistance and AML disease progression[25]. Furthermore, IL6 and ferritin are pro-inflammatory markers strongly associated with poor outcomes in patients with AML, and the anti-inflammatory molecule dexamethasone potentiates the effect of chemotherapies in patients with AML displaying a high white blood cell count[26], demonstrating the urge to find new markers of inflammation to better adapt treatment of AML to the patient. Interestingly, our results demonstrate that low *PRMT2* expression in patients with AML is correlated with higher IL6 and ferritin expressions and is associated with a worse survival rate. Therefore, PRMT2 could be considered as a protein with anti-inflammatory functions and could become a marker of inflammation in this hematological malignancy.

Recent studies highlight the importance of a highly inflammatory BM microenvironment for AML risk stratification. The characterization of the BM immune microenvironment demonstrates distinct remodeling of the BM in response to AML-driven inflammation, highlighting specific B cell

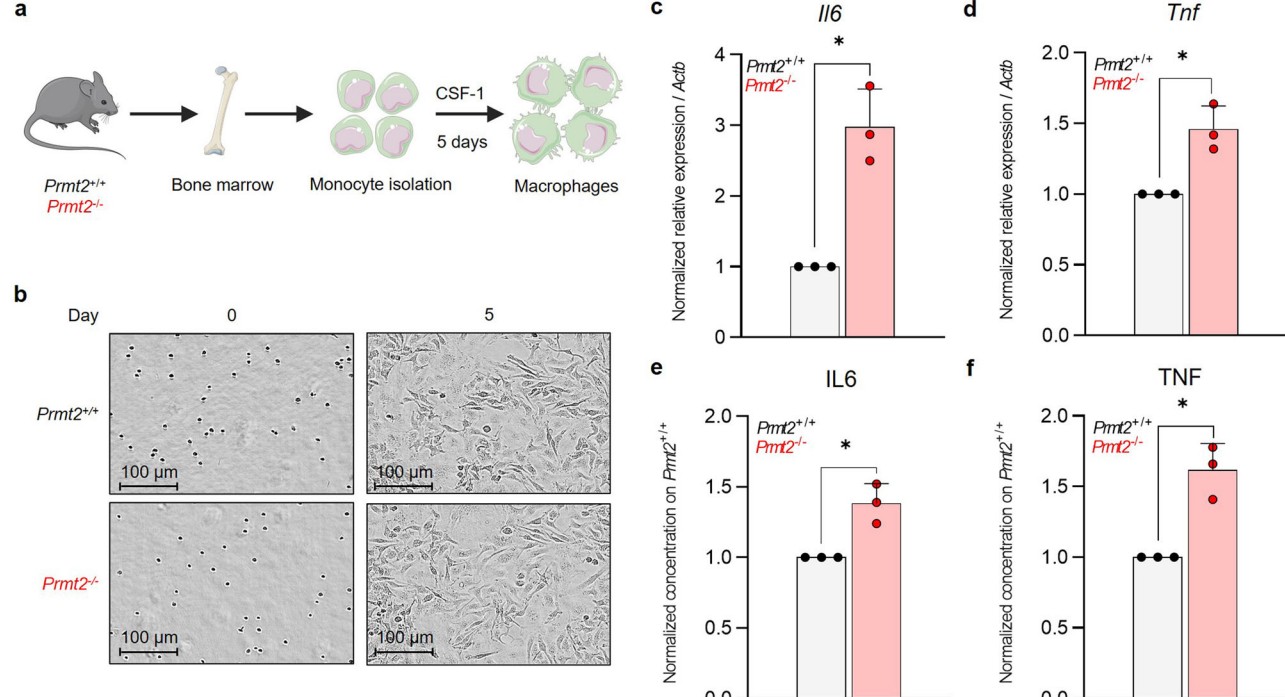

**Fig. 4 | The absence of PRMT2 exacerbates the inflammatory phenotype of mouse bone marrow-derived macrophages. a** Schematic representation for the obtention of BMDMs from purified monocytes from *Prmt2⁻/⁻* or control (*Prmt2⁺/⁺*) mouse BM. **b** Morphological analysis of monocytes after 0 and 5 days of differentiation into macrophages. Images from phase-contrast microscopy using an Incucyte® S3 Live-Cell Analysis System at magnification x20. *Il6* (**c**) and *Tnf* (**d**) expressions measured by RT-QPCR on total RNA from mouse BMDMs after stimulation by 100 ng/mL LPS for 8 h. ELISA on BMDM culture media for IL6 (**e**) and TNF (**f**) after stimulation by 100 ng/mL LPS for 8 h. All error bars represent the mean ± SD of three independent experiments. One sample *t* test with *p* values indicated in the graphs: \**p* < 0.05.

and T-cell populations that are expanded in 'highly inflammatory' patients[27]. Also, cytokine deregulation may contribute to AML development and progression[28]. It has been shown that cytokine signaling pathways are altered in AML, with impacts on prognosis and drug efficiency. Indeed, high concentrations of TNF are associated with an adverse prognosis[29]. IL6 is also considered as a critical cytokine, implicated in regulatory pathways of inflammation, immune regulation[30], and cancer[31,32]. Overexpression of IL6 was detected in patients with AML and has been correlated to a poorer prognosis and drug resistance[33–36].

Arginine methylation, known to control several cellular processes such as transcription, mRNA splicing, and signal transduction, also contributes to the control of inflammation. Although certain roles of PRMTs on inflammatory pathways have been elucidated, no substantial work has uncovered yet the link between PRMTs and inflammation in the context of AML. In this study, we find that in absence of PRMT2 in AML cells, the canonical NF-κB signaling pathway is overactivated, resulting in the induction of its target genes such as *IL6*, promoting the JAK/STAT3 pathway through the binding of IL6 to its receptor. Therefore, PRMT2 could interact with or be part of one of the NF-κB functional subunits to inhibit their functions, either by methylating them or through the methylation of histone tail arginine residues, thus modulating the transcription of NF-κB-related genes. The precise role of PRMT2 on this signaling pathway remains to be uncovered.

However, PRMT2 has already been studied in an inflammatory context. The increased expression and secretion of IL6 and TNF in *Prmt2⁻/⁻* mice have already been described in macrophages[37] and confirm our findings, although we also assess the hematopoietic compartment of this model. It was reported an effect of PRMT2 on the NF-κB signaling pathway using murine *Prmt2⁻/⁻* fibroblasts[38] and it has also been shown that PRMT2 deficiency may enhance the inflammatory pathways in macrophages coming from atherosclerosis plaques[39]. PRMT2 also modulates the alternative splicing of *BCL-X*, another NF-κB target gene, in inflammatory conditions[40].

Conversely, PRMT2-induced arginine methylation of TLR4 cytoplasmic domain has been described as a positive regulation of the IFNβ signaling[41]. In vascular smooth muscle cells, PRMT2 controls the angiotensin II-related proliferation and inflammation[42].

Moreover, some methylation activities of other PRMTs on the NF-κB pathway have previously been described in the literature. Indeed, PRMT1 acts as an inhibitor of inflammation by methylating RelA/p65[43] and the association of PRMT1 with CARM1 co-activates STAT5 for the upregulation of CITED2, resulting in the negative regulation of NF-κB[44]. Conversely, PRMT5 stimulates the NF-κB signaling pathway through an interaction with the TRAIL receptor and methylates RelA/p65, thus promoting pro-inflammatory cytokine expression[45]. PRMT6 and CARM1 also participate in the activation of inflammatory processes by directly acting on RelA/p65 and in the chromatin remodeling processes at the promoter of pro-inflammatory genes[46–48].

Taken together, our results point to a critical role for PRMT2 in the control of inflammation in AML. We have found that PRMT2-depleted AML cells exhibit a pro-inflammatory signature. Of interest, the enriched inflammatory signaling pathways include several inflammation-related shared genes such as *CSF1*, *CXCL10*, *CXCL11* and *IL6*, which could represent potential targets of co-transcriptional activity involving PRMT2. This would be of particular interest for further investigations on the role of PRMT2 at the chromatin level. Also, *Prmt2⁻/⁻* murine BMDMs stimulated with LPS display increased expression and secretion of the IL6 and TNF pro-inflammatory cytokines, indicating that PRMT2 is involved in the inflammatory response in normal hematopoiesis. Furthermore, we demonstrate that PRMT2-depleted AML cells exhibit heightened activation of the NF-κB signaling pathway, leading to IL6 overexpression, and ultimately to an increased phosphorylation of the STAT3 protein. The other signaling pathways giving rise to activation of the JAK/STAT3 signal are not impacted by PRMT2. Finally, in keeping with our mechanistic work, we noticed that adult patients with AML harboring a low expression of *PRMT2* display a

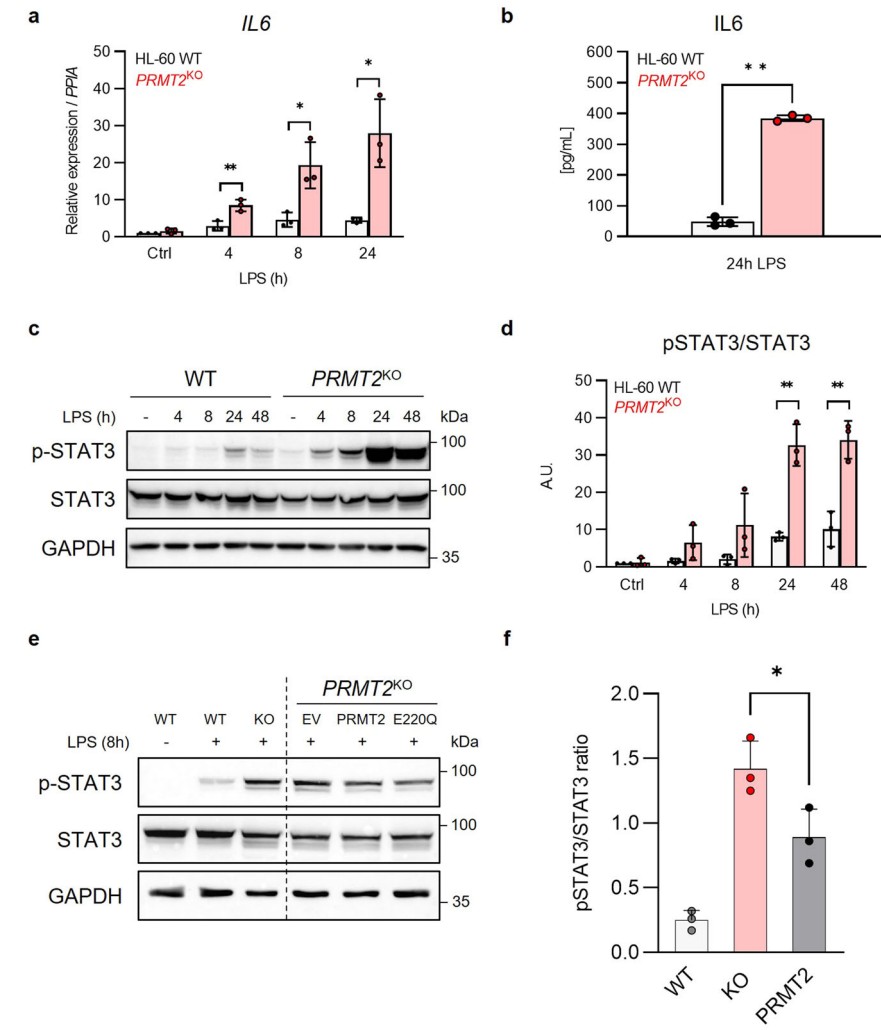

**Fig. 5 | PRMT2 is involved in the human inflammatory response through the control of STAT3 activation in vitro upon stress. a** RT-QPCR on total RNA measuring *IL6* in *PRMT2*KO and WT cells after stimulation by 100 ng/mL LPS during 4, 8, or 24 h. **b** ELISA on human IL6 in *PRMT2*KO and WT cell culture medium after stimulation by LPS (100 ng/mL) during 24 h. Western Blot analysis (**c**) and quantitation (**d**) of phosphorylated (Y705) and total forms of STAT3 after stimulation by LPS (100 ng/mL) during 4, 8, 24 or 48 h. GAPDH is used as a loading control. Western Blot analysis (**e**) and quantitation (**f**) of phosphorylated and total forms of STAT3 after stimulation by LPS (100 ng/mL) during 8 h of HL-60 WT, KO, or KO cells infected with viruses containing DNA coding for either an empty vector (EV), exogenous PRMT2 (PRMT2) or a catalytically dead mutant (E220Q). GAPDH is used as a loading control. All the presented blots are representative figures from at least three independent experiments. All error bars represent the mean ± SD of three independent experiments. Ctrl: Control. One-way ANOVA (**a**, **d**, **f**) or one sample *t* test (**b**) with *p* values indicated in the graphs: *$p < 0.05$; **$p < 0.01$.

lower survival rate and an increased activation of inflammatory signatures compared to the high *PRMT2* expressors. Although the mechanisms are not yet fully understood at the molecular level, we provide here an assessment of the close link between PRMTs and the inflammatory pathways in acute myeloid leukemia. Our study allows a better understanding of the clinical relevance of using PRMT2 as a novel marker of inflammation, thus helping in the prediction of the patient response to anti-inflammatory drugs in the treatment of AML.

## Methods

### Plasmids, transfection, and retrovirus production

The Myc-MOZ vector was provided by Edward Chan (Indiana University Cancer Center, Indianapolis, IN, USA). The pMA-citrine-PRMT2 plasmid was designed and purchased from Thermofisher Scientific and was subcloned into a pMSCV-pgk-puro (Addgene #68469) vector. For Myc-MOZ and citrine-PRMT2 co-expression assay, HEK293 cells at 80% confluence were transfected with the Myc-MOZ vector (3.5 µg) and the pMSCV-citrine-PRMT2-pgk-puro plasmid (3.5 µg), using 20 µL of a non-liposomal reagent, following the manufacturer's protocol (JetPRIME, Polypus transfection). Cells were then incubated for 24 h at 37 °C. For retrovirus production, HEK293 were transfected as previously described with 1 µg of Gag-pol, 1 µg of VSV-G, and 8 µg of pMSCV-citrine-PRMT2-pgk-puro or pMSCV-PRMT2-E220Q-pgk-puro plasmids. The medium was removed after 4 h, washed with 1X PBS (Dutscher) and replaced. After 24 h, the first virus-containing medium was trashed and replaced with fresh medium. Next supernatants were harvested for 4 days and kept at -80 °C.

### Retrovirus cell infection

The day prior infection, wells were coated with 10 µg RetroNectin (Takara bio) per cm² and incubated overnight at 4 °C. RetroNectin was removed, washed twice with 1X PBS and viruses were added. The plate was then centrifuged at maximum speed (around 3000 × *g*) for 1 h at RT with the lid. After centrifugation, the supernatant was removed, rinsed with 1X PBS and the HL-60 cells were added at a concentration of 1 million/mL. They were incubated for 48 h at 37 °C, centrifuged, and washed twice with 1X PBS to remove remaining viruses. Infection efficiency was assessed by detecting the citrine-PRMT2 protein by flow cytometry.

### CRISPR-Cas9 nucleofection

Two crRNA targeting PRMT2 were designed and purchased from Integrated DNA Technologies: AA (5'-CCUGACGGAUAAAGU-3') and AB (Forward: 5'-CGUGGAUGAGUACGA-3'). The ribonucleoprotein complexes were formed as specified in the manufacturer's protocol (Alt-R CRISPR-Cas9 System, Integrated DNA Technologies). They were further electroporated into HL-60 cells through the Amaxa Nucleofector System (Amaxa Cell Line Nucleofector Kit V, Lonza) following the manufacturer's protocol. After 48 h, cells were washed and resuspended in Pre-Sort Buffer (BD Biosciences) and a single-cell sorting was performed in a 96-well plate containing culture medium supplemented with 50% of HL-60 conditioned medium using a BD FACSMelody (BD Biosciences). The *PRMT2*KO was then assessed for each subclone by DNA sequencing, RT-qPCR, and Western Blotting.

**Fig. 6 | The absence of PRMT2 deregulates the NF-κB signaling pathway in AML upon stress.** Western Blot analysis (**a**) and quantitation (**b**) of phosphorylated (Y705) and total forms of STAT3 after stimulation by IL6 (40 ng/mL) during 0.5, 2, 8 or 24 h, shown as mean ± SD of three independent experiments and analyzed by one-way ANOVA. **c** Western Blot analysis of phosphorylated and total forms of the SAPK/JNK, ERK1/2, and p38 MAPKs on *PRMT2*^KO and WT cells after stimulation by 100 ng/mL LPS during 4, 8, 24 or 48 h. **d** Western Blot analysis of NF-κB p65 and phosphorylated (Y705) STAT3 nuclear translocation after cellular fractionation of *PRMT2*^KO and WT HL-60 cells. SP1 and ACTB correspond to nuclear and cytoplasmic controls, respectively. Ctrl: Control. All presented blots are representative figures from three independent experiments.

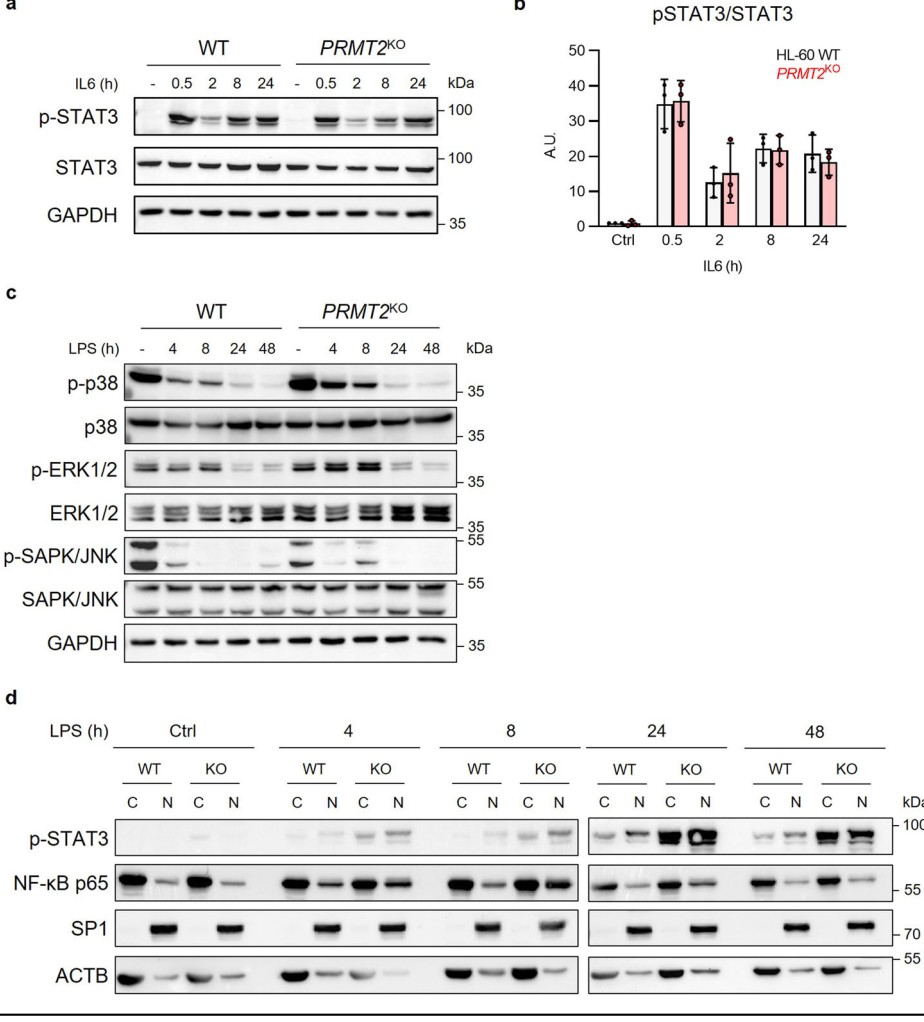

## Immunoprecipitation

For immunoprecipitation, we followed protocols provided by the manufacturer (GFP-Trap Magnetic Particles M-270 and Myc-Trap Magnetic Agarose, Chromotek) with some adjusting to our needs. Briefly, $5 \times 10^6$ cells were lysed on ice for 30 min in 200 μL ice-cold lysis buffer (10 mM Tris/HCl pH 7.5, 150 mM NaCl, 0.5 mM EDTA, 0.5% NP40) supplemented with a protease inhibitor cocktail (cOmplete, Sigma-Aldrich) and a phosphatase inhibitor cocktail (PhosSTOP, Sigma-Aldrich) with frequent homogenization. The lysates were centrifuged at 17,000×g for 10 min at 4 °C. Some of the collected supernatants were used as input fractions. The remaining supernatants were incubated with 25 μL of GFP-Trap Magnetic particles beads and rotated end-over-end for 1 h at 4 °C. Beads were separated using a magnet until supernatant was clear. Supernatants were discarded and 50 μL was stored as the unbound fraction. The bead-bound materials were washed four times with 500 μL of wash buffer (10 mM Tris/HCl pH 7.5, 150 mM NaCl, 0.05% NP40, 0.5 mM EDTA). The beads were eluted with 80 μL of 2X Laemmli and boiled at 95 °C for 5 min and the beads were discarded from the proteins using a magnet.

## Western Blot

Total cell lysates were obtained by resuspending cells in 1X cell lysis buffer (10X stock solution: 9803, Cell Signaling) supplemented with protease inhibitor cocktail and phosphatase inhibitor cocktail as previously described. They were further incubated 30 min on ice with frequent homogenization and then centrifuged at 17,000×g for 15 min. NE-PER™ Nuclear and Cytoplasmic Extraction kit (78833, Thermo Scientific) was used for cell fractionation following the manufacturer's protocol. To prepare samples for Western Blot analyses, 50 μg of the protein supernatant were taken and mixed with Laemmli buffer 1X (4X stock solution: Tris-HCl pH 6.8 200 mM, DTT 400 mM, SDS 277 mM, bromophenol blue, glycerol 30%). Samples were then heated at 95 °C for 5 min. Afterward, samples were separated by SDS-PAGE (4% stacking gel, 10% resolving gel) and transferred into a nitrocellulose membrane (Bio-Rad). The membranes were blocked for 1 h in BSA (5% solution in 1X TBS-T (0.1%)) and stained sequentially with primary and secondary antibodies. The following primary antibodies were used: Anti-mouse Myc-tag (60003-2-Ig, Proteintech, at 1/5000th), PRMT4 (12495), GAPDH (97166) and anti-rabbit GFP (2555), PRMT1 (ab190892, Abcam), PRMT2 (LS-C482512, LS-Bio), PRMT3 (ab191562, Abcam), PRMT5 (79998), p-STAT3 (9145), STAT3 (12640), p-IκBα (2859), IκBα (4812), p-SAPK/JNK (9251), SAPK/JNK (9252), p-ERK1/2 (9101), ERK1/2 (9102), p-p38 (4511), p38 (9212), SP1 (9389), p65 (8242). An anti-beta actin directly coupled with HRP (A3854, Sigma, at 1/50000th) was also used. All antibodies were purchased from Cell Signaling and were all diluted at 1/1000th (except when specified) in blocking buffer and used at 4 °C overnight, then washed three times in 1X TBS-T (0.1%) during 10 min each. The secondary anti-rabbit (7074) or anti-mouse (7076) antibody conjugated with horseradish peroxidase was added, and membranes were incubated at RT for 1 h. Membranes were then washed three times in 1X TBS-T (0.1%) during 10 min each. To obtain the chemiluminescence signal, membranes were incubated 5 min with Clarity or Clarity Max Western ECL Substrate (Bio-Rad) and read on a ChemiDoc MP Imaging System (Bio-Rad).

## Patients with AML cohorts

Reanalysis of previously obtained RNA sequencing, clinical and survival data from the Leucegene Project and the BeatAML cohorts were processed. Availability of these public datasets is described in 'Data Availability Statement'.

## Mice

We have complied with all relevant ethical regulations for animal use. Experiments using mice were approved by the Ethics committee of the Université de Bourgogne, which is an Institutional Animal Care and Use Committee. Mice were housed in a temperature-controlled environment under a 12 h light-dark cycle with free access to water and a standard irradiated rodent chow diet. All mice used were maintained under specific pathogen-free conditions according to animal-study protocols reviewed and approved in accordance with Université de Bourgogne institutional guidelines for animal care and under protocols approved by the Université de Bourgogne Institutional Animal Care and Use Committee. $Prmt2^{-/-}$ mice were generated by Elizabeth Nabel (Cardiovascular Branch, National Heart, Lung, and Blood Institute, National Institutes of Health, Bethesda, MD, USA) and provided by Yann Hérault. $Prmt2^{-/-}$ mice were backcrossed ten times with C57BL/6JRj mice (Janvier Lab) to ensure a pure genetic background. For genotyping, genomic DNA was prepared from tail biopsies using the REDExtract-N-Amp™ Tissue PCR Kit (Sigma-Aldrich). Deleted (280 bp) and wild type (190 bp) $Prmt2$ alleles were identified by PCR with three specific primers: A (5′-CTGAGGTATTACCAGCAGACA-3′), B (5′-CTCTCTGATGCAGGTCTAC-3′) and C (5′-CCGGTGGATGTGGA ATGTGT-3′).

## BM cell preparation

BM cells were isolated by crushing mice femora and tibiae using 1X PBS supplemented with 2% FBS and filtered through a 70 μm mesh. Blood from control and $Prmt2^{-/-}$ mice was collected from the lateral tail vein using EDTA-coated tubes (BD Microtainer, BD Biosciences). Complete blood counts were performed using an automated hematology analyzer (Scil Vet abc Plus + , Scil animal care company).

## BMDM differentiation and treatment

BMDMs were obtained from 6- to 8-week-old male/female mice and monocytes were magnetically isolated from total BM filtered through a 45 μm mesh (Monocyte Isolation Kit (BM), mouse, Miltenyi). They were cultured with 100 ng/mL murine CSF-1 (130-101-706, Miltenyi) for five days. An inflammatory phenotype was induced after five days of differentiation by adding 100 ng/mL of sonicated LPS (E. coli O127:B8, Sigma-Aldrich) to the culture medium. Pictures of the macrophages were taken every 12 h during differentiation using a live imager (Incucyte® S3 Live-Cell Analysis System, Sartorius).

## BM competitive reconstitution

$Prmt2^{-/-}$, $Prmt2^{+/+}$ and $Ly5.1^+$ ($CD45.1^+$) mice were euthanized and BMs were harvested as previously described. Five BM cell preparations were made at a concentration of $4 \times 10^6$/mL: only $Ly5.1^+$ cells; only $Prmt2^{-/-}$ cells; only $Prmt2^{+/+}$ cells; half $Ly5.1^+$, half $Prmt2^{-/-}$; half $Ly5.1^+$, half $Prmt2^{+/+}$ cells. For each preparation, five sub-lethally irradiated (9 Gy) $Ly5.1^+$ mice were transplanted with 0.8 million cells. Every two weeks after the transplantation, three drops of blood were collected from the tail vein in an EDTA-coated tube (BD Microtainer with K2EDTA, BD Biosciences). Red blood cells were lysed (10X RBC lysis buffer: 155 mM $NH_4Cl$, 12 mM $NaHCO_3$, 0.1 mM EDTA) 10 min at 4 °C and cells were centrifuged at 300 g for 5 min.

## Cell culture and treatments

The human AML cell line HL-60 obtained from the American Type Culture Collection (ATCC) and subclones and the murine BMDMs were both grown in RPMI 1640 with L-Glutamine (Dutscher) supplemented with 10% FBS (Dutscher) and 1% of antibiotics (10,000 U/mL Penicillin, 10 mg/mL

Streptomycin, 25 μg/mL Amphotericin B, PAN-Biotech). The human cell line HEK293 obtained from the ATCC and subclones were grown in DMEM (Dutscher) supplemented with 10% FBS and 1% of antibiotics as previously described. All cells were incubated in a humified atmosphere with 5% $CO_2$ at 37 °C. For inflammatory phenotype experiments, HL-60 cells were treated with 100 ng/mL sonicated LPS (E. coli O127:B8, Sigma-Aldrich) or 40 ng/mL human IL6 (SRP3096, Merck). For proliferation assays, cells were diluted in 1/5$^{th}$ with Trypan Blue (Gibco) and counted on Malassez chamber.

## Flow cytometry

Cells were centrifuged at $300 \times g$ for 5 min and washed twice with 1X PBS. Antibodies are then added and diluted in 1X PBS supplemented with 2% FBS and incubated for 20 min at 4 °C. After incubation, excess antibodies were diluted in 1X PBS supplemented with 2% FBS, samples were centrifuged at $300 \times g$ for 5 min and washed twice with 1X PBS. Sample acquisition was performed using a LSR Fortessa or a LSR II (BD Biosciences) flow cytometer and data were analyzed with FlowJo (v10.5.3).

Anti-human CD126-PE (BD Biosciences, 561696) and anti-mouse Lineage-APC (BD Biosciences, 558074), c-Kit-BV510 (BioLegend, 135119), Sca-1-PE-Cy7 antibody (eBioscience, 25-5981-82), CD11b-PE-Cy7 (BD Biosciences, 552850), Gr-1-FITC (BD Biosciences, 553127), Ter119-V500 (BD Biosciences, 562120), B220-AF700 (BioLegend, 103232), and CD71-APC antibodies (BioLegend, 113820), CD80-APC-Vio770 (Miltenyi, 130-116-463), CD14-PE-Cy7 (BioLegend, 123316), CD16-FITC (BD Biosciences, 553144), CD64-PE (Miltenyi, 130-103-808), CD206-APC (BioLegend, 141708), F4/80-BV421 (BD Biosciences, 565411), Ly5.1-eFluor450 (Invitrogen, 48-0453-82), and Ly5.2-APC (BioLegend, 109814) antibodies were all used at 1/300$^{th}$ dilution.

Analysis of the HL-60 immunophenotype was performed by the flow cytometry platform of the Department of Hematology Biology (University Hospital Dijon Bourgogne François-Mitterrand). Cell samples were processed in a "lysis-no wash" protocol. Sample acquisition was performed using a Navios® (Beckman Coulter) flow cytometer. Following antibodies were used: CD15-FITC (Beckman Coulter, B36298, at 1/10$^{th}$), CD14-PE (Beckman Coulter, A07764, at 1/10$^{th}$), CD13-PE-DyLight™594 (SYSMEX, BM509936, at 1/50$^{th}$), CD33-PC5.5 (Beckman Coulter, B36289, at 1/20$^{th}$), CD34-PC7 (SYSMEX, CU114497, at 1/50$^{th}$), CD117-APC (SYSMEX, CE001108, at 1/20$^{th}$), CD11b-APC-A750 (Beckman Coulter, B36295, at 1/50$^{th}$), CD16 BV421 (BD Biosciences, 562874, at 1/40$^{th}$). Gating strategies for all flow cytometry data are provided in the Supplementary Fig. 6a–c.

## RNA isolation and real-time quantitative PCR

Total RNA was extracted with the RNeasy Mini Kit (Qiagen) with a previous step to homogenize the samples (QIAshredder, Qiagen). cDNA was synthesized from 1 μg of RNA using the sensiFAST cDNA Synthesis Kit (Bioline) following the manufacturer's instructions. cDNA was amplified with the GoTaq Probe qPCR Master Mix (Promega) using the ViiA™ 7 Real-Time PCR System (Applied Biosystems). Mouse Taqman probes for $Il6$ (Mm00446190_m1) and $Tnf$ (Mm00443250_m1) and human Taqman probes for $IL6$ (Hs00174131_m1) was used. Mouse $B$-$Actin$ (Mm02619580_g1) and human $PPIA$ (Hs04194521_s1) were used as normalization controls. All probes were purchased from Thermofisher Scientific.

## ELISA

All ELISA kits were purchased from BD Biosciences and performed following the manufacturer's protocol. For human samples, the Human IL6 ELISA Kit II (550799) was used and for murine samples the BD OptEIA™ Mouse IL6 ELISA (550950) and the BD OptEIA™ Mouse TNF ELISA (560478) kits were used.

## RNA-Seq and GSEA

For high-throughput RNA-seq, 1 million cells were harvested from HL-60 wild type, HL-60 $PRMT2^{KO}$ or the same cell lines treated with 100 ng/mL

LPS during 24 h. RNA was extracted (TRIzol™ reagent, Invitrogen) and eluted in 15 μl of RNAse-free water. RNA-seq and library preparation were performed by GENEWIZ (Azenta Life Sciences). The paired-end reads were aligned to the human genome reference GRCh38 (genome-build-accession GCA_000001405.28) with the well-known aligner STAR (v2.7.6a)[49] using the 2-pass mode. The read counts were also done with STAR. The library DESeq2 (v1.39.8)[50] from R (v4.1.1) was used to normalize the read counts and perform the differential gene expression analysis based on the negative binomial distribution.

## Statistics and reproducibility
Overall survival was determined from the time of diagnosis until death or last follow-up. Survival curves were calculated using the Kaplan-Meier method and compared using the Log-rank test. Normality of data was assessed with the Shapiro-Wilk test. For normally distributed data, an unpaired $t$ test was performed, otherwise the non-parametric Mann-Whitney test was used. One-way ANOVA was used to analyze the difference between the means of more than two groups. All graphs and statistical analyses were performed with GraphPad Prism (v9.5.1).

## Reporting summary
Further information on research design is available in the Nature Portfolio Reporting Summary linked to this article.

## Data availability
The Leucegene Project RNA sequencing data have been previously deposited in the Gene Expression Omnibus (accession numbers GSE49642 and GSE52656). Other data from The Leucegene Project are available on reasonable request from Guy Sauvageau. The BeatAML cohort RNA sequencing data are available in the Supplementary Information of the corresponding publication by Tyner et al.[51]. HL-60 RNA sequencing data have been deposited by the authors on Gene Expression Omnibus (accession number GSE266252). All source data are shown in the 'Supplementary Data 1' file. All unedited Western Blot images are displayed in Supplementary Fig. 7. All other data are available from the corresponding author on reasonable request.

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

## Acknowledgements

We gratefully acknowledge the IMAFLOW Core Facility at the Université de Bourgogne supported by Burgundy Regional Council (Anabelle Sequeira, Serge Monier, and Nicolas Pernet) for all the flow cytometry data acquisition and helpful advice on data analyzes, and the animal facility (Valérie Saint-Giorgio) from the Université de Bourgogne for the mouse studies. We also thank Dr François Hermetet for providing the antibodies against phosphorylated and total MAPK proteins. This work was supported by a French Government grant managed by the French National Research Agency (ANR) under the program "Investissements d'Avenir" with reference ANR-11 LABX-0021 (LipSTIC LabEx), the Association Laurette Fugain (to Laurent Delva), the Ligue Nationale contre le Cancer (Coordination Interrégionale des régions Grand Est et Bourgogne Franche Comté) (to Laurent Delva), the Cancéropôle Est (to Laurent Delva), the FEDER, and the Regional Council of Bourgogne Franche Comté. Camille Sauter was supported by fellowships from the Fondation pour la Recherche Médicale (FRM) (ECO201906009006; 3 years), the Fondation ARC (ARCDOC42022010004491; 1 year) and the Société Française d'Hématologie (SFH; 4 months), John Simonet by the LipSTIC LabEx, Denis Masnikov and Baptiste Pernon by the French Ministère de l'Enseignement Supérieur, de la Recherche et de l'Innovation (MESRI), and Dr. Anne Largeot by fellowships from Inserm associated with the Regional Council of Bourgogne and the SFH.

## Author contributions

C.S. contributed to research design, performed the research, analyzed the data, and drafted the manuscript; T.M. contributed to experiments, data analysis, and discussion; F.G. contributed to data analysis, helpful discussions, and critically reviewed the manuscript; J.S. conducted experiments and contributed to data analysis; C.F. analyzed omic data; C.R. performed and analyzed flow cytometry data; D.M. participated to data analysis, and critically reviewed the manuscript; B.P. critically reviewed the manuscript; A.L. conducted yeast two-hybrid assay; A.A. performed experiments; Y.H. and G.S. provided study materials; M.M. critically reviewed the manuscript; M.C. designed and co-interpreted the patients with AML RNA-seq/GSEA, exome and ImmuCellAI analyses; J.N.B. participated in the research design, contributed to discussions and critically reviewed the manuscript; R.A. and L.D. contributed to research design, supervised the research and critically reviewed the manuscript.

## Competing interests

The authors declare no competing interests.
