## [Peer review file · Communications Biology]

Referee expertise:

Referee #1: Arginine methylation and cancer, PMTs, cell signaling

Referee #2: Epigenetics, drug discovery

Reviewers' comments:

Reviewer #1 (Remarks to the Author):

This manuscript investigated the role of PRMT2 in AML. The results showed that AML patients with low expression of PRMT2 have lower survival rate, higher expression of TNF/NFκB signaling and inflammation signature. The authors further validated their findings in both PRMT2 knockout HL-60 cells and BM-derived macrophages of Prmt2 knockout mice, as well as showed that knockout of PRMT2 activates STAT3 likely through NFκB signaling.

Although the results support the major conclusion, it did not provide novel knowledge on the role of PRMT2 in NFκB and inflammatory signaling, which has been demonstrated by multiple studies as mentioned in the Discussion. It is also unclear whether PRMT2 plays any role in AML development.

Reviewer #2 (Remarks to the Author):

This manuscript reported the role of PRMT2 in AML. Through expression screening of an AML patient cohort and analysis of genetic method, authors discovered that PRMT2 as a key regulator of inflammation in AML through NF-κB-STAT3-IL6 axis. The discovery is novel. However, the study would be strengthened with detailed mechanistic studies to elucidate the role of PRMT2 in this axis. Below are some concerns and suggestions.

1. Does PRMT2 regulate STAT3 phosphorylation level through its catalytic function or scaffold function? If through catalytic function, what is the substrate?
2. In Fig 3, is there any shared genes in TNFA signaling, inflammatory and IL6-JAK-STAT6 hallmarks (in Fig d)?

3. Minor: combine a & b together in Fig 3.

Point-by-point response to reviewers

Reviewer #1 (Remarks to the Author):

This manuscript investigated the role of PRMT2 in AML. The results showed that AML patients with low expression of PRMT2 have lower survival rate, higher expression of TNF/NFκB signaling and inflammation signature. The authors further validated their findings in both PRMT2 knockout HL-60 cells and BM-derived macrophages of Prmt2 knockout mice, as well as showed that knockout of PRMT2 activates STAT3 likely through NFκB signaling.

1. Although the results support the major conclusion, it did not provide novel knowledge on the role of PRMT2 in NFκB and inflammatory signaling, which has been demonstrated by multiple studies as mentioned in the Discussion.

While the precise molecular mechanisms remain partially elucidated, our study represents a pioneering exploration of the intricate interplay between PRMTs and inflammatory pathways in acute myeloid leukemia (AML). Our focus is made on understanding the clinical implications of PRMT2 as a potential inflammation marker, thereby enhancing prognostic accuracy for patient response to anti-inflammatory therapeutics in AML treatment.

The novelty of our work also resides in the models used, through the examination of the hematopoietic compartment of *Prmt2*^{-/-} mice, which has never been done previously. Additionally, we conducted transcriptomic analyses comparing PRMT2 knockout (*PRMT2*^{KO}) and wildtype (WT) AML cell lines, a novel investigative angle that enriches our understanding of PRMT2's role in AML.

2. It is also unclear whether PRMT2 plays any role in AML development.

Recent studies highlighted the importance of a highly inflammatory bone marrow (BM) microenvironment for AML risk stratification. The characterization of the BM immune microenvironment demonstrates distinct remodeling of the BM in response to AML-driven inflammation, highlighting specific B cell and T-cell populations that are expanded in 'highly inflammatory' patients (Lasry *et al.*, Nat Cancer, 2023). By modulating the inflammatory response, PRMT2 could contribute to the development of AML and influence therapeutic responses.

We thank reviewer #1 for her/his insightful comments. In response, we have meticulously addressed and expanded upon the pertinent points raised in the Discussion section. This includes the addition of new sentences and incorporation of relevant bibliographical references to enhance the clarity and comprehensiveness of our findings (**lines 313-316 and lines 366-370**).

Lines numbers are given from the revised article file, with the 'all markup' setting.

Reviewer #2 (Remarks to the Author):

This manuscript reported the role of PRMT2 in AML. Through expression screening of an AML patient cohort and analysis of genetic method, authors discovered that PRMT2 as a key regulator of inflammation in AML through NF- κ B-STAT3-IL6 axis. The discovery is novel. However, the study would be strengthened with detailed mechanistic studies to elucidate the role of PRMT2 in this axis. Below are some concerns and suggestions.

We thank reviewer #2 for her/his positive comments.

1. Does PRMT2 regulate STAT3 phosphorylation level through its catalytic function or scaffold function? If through catalytic function, what is the substrate?

We investigated the mechanism by which PRMT2 influences STAT3 phosphorylation. To answer reviewer question, we used the PRMT2^{KO} HL-60 model in which we rescued PRMT2 expression by transducing the cells with a retroviral vector carrying either the wildtype PRMT2 or a catalytically inactive mutant. This mutant, designated as 'E220Q', has been engineered by replacing the glutamic acid residue at position 220 with glutamine (see figure below). This catalytically dead mutant has already been used in previous studies (Pak *et al.*, Biochemistry, 2011; Vhuyian *et al.*, J Biochem, 2017).

We treated WT, PRMT2^{KO}, and KO cells rescued with either an empty vector (EV), WT PRMT2, or the E220Q PRMT2 mutant with LPS, and subsequently assessed the level of phosphorylated STAT3. As stated in the manuscript, we observed a reduction in the pSTAT3/STAT3 ratio in the PRMT2-rescued cells compared to KO cells.

Of particular interest, we found that the pSTAT3/STAT3 ratio was equivalent in the PRMT2 and E220Q-rescued cells, indicating that the phosphorylation of STAT3 is not impacted by the loss of catalytic activity of PRMT2.

As shown in the manuscript (Supplemental Figure 5b), the influence of PRMT2 on STAT3 phosphorylation does not appear to arise from a direct interaction between these two proteins, as evidenced by the absence of detectable signal following co-immunoprecipitation.

Taken together, these findings suggest that STAT3 is unlikely to be a substrate of PRMT2 in AML cells. Therefore, the modulation of STAT3 activity may arise from the scaffolding function of PRMT2. Consequently, further investigations through interactome analyses could provide insights into new targets of PRMT2 in AML.

Figure 5e was modified in the article manuscript so that the E220Q band is shown (**new Figure 5 image below**) as well as the corresponding legend (**line 269**). A text was added to further describe this experiment and our conclusions (**lines 246-249 and lines 255-257**).

2. In Fig 3, is there any shared genes in TNFA signaling, inflammatory and IL6-JAK-STAT6 hallmarks (in Fig d)?

Here we present the 9 genes shared among the three different gene sets: 'TNFA_VIA_NFKB' [200 genes], 'INFLAMMATORY_R' [200 genes] and 'IL6_JAK_STAT3' [87 genes]: CSF1, CXCL10, CXCL11, IFNGR2, IL15RA, IL1B, IL6, IRF1, and TLR2.

Hence, these genes could be viewed as potential targets of co-transcriptional activity involving PRMT2 and would be of particular interest for further investigations on the role of PRMT2 on chromatin regulation and gene expression.

The inclusion of this point in the discussion section of the revised manuscript appropriately highlights the significance of these genes and the potential co-transcriptional activity of PRMT2 (lines 352-356).

3. Minor: combine a & b together in Fig 3.

After incorporating the requested changes, Figure 3a and b have been merged under the designation "Figure 3a". Additionally, subsequent figure parts have been renumbered accordingly (**new Figure 3 image below**). The figure legend has been revised (**lines 183-193**) to accurately reflect the changes, along with corresponding adjustments made to the text (**lines 166-179**).

REVIEWERS' COMMENTS:

Reviewer #1 (Remarks to the Author):

The explanations to my comments are acceptable. The new information in the Discussion section is helpful.

Reviewer #2 (Remarks to the Author):

Authors addressed the majority of comments and the manuscript quality has been improved.

Point-by-point response to reviewers

Reviewer #1 (Remarks to the Author):

This manuscript investigated the role of PRMT2 in AML. The results showed that AML patients with low expression of PRMT2 have lower survival rate, higher expression of TNF/NFκB signaling and inflammation signature. The authors further validated their findings in both PRMT2 knockout HL-60 cells and BM-derived macrophages of Prmt2 knockout mice, as well as showed that knockout of PRMT2 activates STAT3 likely through NFκB signaling.

1. Although the results support the major conclusion, it did not provide novel knowledge on the role of PRMT2 in NFκB and inflammatory signaling, which has been demonstrated by multiple studies as mentioned in the Discussion.

While the precise molecular mechanisms remain partially elucidated, our study represents a pioneering exploration of the intricate interplay between PRMTs and inflammatory pathways in acute myeloid leukemia (AML). Our focus is made on understanding the clinical implications of PRMT2 as a potential inflammation marker, thereby enhancing prognostic accuracy for patient response to anti-inflammatory therapeutics in AML treatment.

The novelty of our work also resides in the models used, through the examination of the hematopoietic compartment of *Prmt2*^{-/-} mice, which has never been done previously. Additionally, we conducted transcriptomic analyses comparing PRMT2 knockout (*PRMT2*^{KO}) and wildtype (WT) AML cell lines, a novel investigative angle that enriches our understanding of PRMT2's role in AML.

2. It is also unclear whether PRMT2 plays any role in AML development.

Recent studies highlighted the importance of a highly inflammatory bone marrow (BM) microenvironment for AML risk stratification. The characterization of the BM immune microenvironment demonstrates distinct remodeling of the BM in response to AML-driven inflammation, highlighting specific B cell and T-cell populations that are expanded in 'highly inflammatory' patients (Lasry *et al.*, Nat Cancer, 2023). By modulating the inflammatory response, PRMT2 could contribute to the development of AML and influence therapeutic responses.

We thank reviewer #1 for her/his insightful comments. In response, we have meticulously addressed and expanded upon the pertinent points raised in the Discussion section. This includes the addition of new sentences and incorporation of relevant bibliographical references to enhance the clarity and comprehensiveness of our findings (**lines 313-316 and lines 366-370**).

Lines numbers are given from the revised article file, with the 'all markup' setting.

Reviewer #2 (Remarks to the Author):

This manuscript reported the role of PRMT2 in AML. Through expression screening of an AML patient cohort and analysis of genetic method, authors discovered that PRMT2 as a key regulator of inflammation in AML through NF- κ B-STAT3-IL6 axis. The discovery is novel. However, the study would be strengthened with detailed mechanistic studies to elucidate the role of PRMT2 in this axis. Below are some concerns and suggestions.

We thank reviewer #2 for her/his positive comments.

1. Does PRMT2 regulate STAT3 phosphorylation level through its catalytic function or scaffold function? If through catalytic function, what is the substrate?

We investigated the mechanism by which PRMT2 influences STAT3 phosphorylation. To answer reviewer question, we used the PRMT2^{KO} HL-60 model in which we rescued PRMT2 expression by transducing the cells with a retroviral vector carrying either the wildtype PRMT2 or a catalytically inactive mutant. This mutant, designated as 'E220Q', has been engineered by replacing the glutamic acid residue at position 220 with glutamine (see figure below). This catalytically dead mutant has already been used in previous studies (Pak *et al.*, Biochemistry, 2011; Vhuyian *et al.*, J Biochem, 2017).

We treated WT, PRMT2^{KO}, and KO cells rescued with either an empty vector (EV), WT PRMT2, or the E220Q PRMT2 mutant with LPS, and subsequently assessed the level of phosphorylated STAT3. As stated in the manuscript, we observed a reduction in the pSTAT3/STAT3 ratio in the PRMT2-rescued cells compared to KO cells.

Of particular interest, we found that the pSTAT3/STAT3 ratio was equivalent in the PRMT2 and E220Q-rescued cells, indicating that the phosphorylation of STAT3 is not impacted by the loss of catalytic activity of PRMT2.

As shown in the manuscript (Supplemental Figure 5b), the influence of PRMT2 on STAT3 phosphorylation does not appear to arise from a direct interaction between these two proteins, as evidenced by the absence of detectable signal following co-immunoprecipitation.

Taken together, these findings suggest that STAT3 is unlikely to be a substrate of PRMT2 in AML cells. Therefore, the modulation of STAT3 activity may arise from the scaffolding function of PRMT2. Consequently, further investigations through interactome analyses could provide insights into new targets of PRMT2 in AML.

Figure 5e was modified in the article manuscript so that the E220Q band is shown (**new Figure 5 image below**) as well as the corresponding legend (**line 269**). A text was added to further describe this experiment and our conclusions (**lines 246-249 and lines 255-257**).

2. In Fig 3, is there any shared genes in TNFA signaling, inflammatory and IL6-JAK-STAT6 hallmarks (in Fig d)?

Here we present the 9 genes shared among the three different gene sets: 'TNFA_VIA_NFKB' [200 genes], 'INFLAMMATORY_R' [200 genes] and 'IL6_JAK_STAT3' [87 genes]: CSF1, CXCL10, CXCL11, IFNGR2, IL15RA, IL1B, IL6, IRF1, and TLR2.

Hence, these genes could be viewed as potential targets of co-transcriptional activity involving PRMT2 and would be of particular interest for further investigations on the role of PRMT2 on chromatin regulation and gene expression.

The inclusion of this point in the discussion section of the revised manuscript appropriately highlights the significance of these genes and the potential co-transcriptional activity of PRMT2 (lines 352-356).

3. Minor: combine a & b together in Fig 3.

After incorporating the requested changes, Figure 3a and b have been merged under the designation "Figure 3a". Additionally, subsequent figure parts have been renumbered accordingly (**new Figure 3 image below**). The figure legend has been revised (**lines 183-193**) to accurately reflect the changes, along with corresponding adjustments made to the text (**lines 166-179**).